# Using Domain Knowledge for Interpretable and Competitive Multi-Class Human Activity Recognition

**DOI:** 10.3390/s20041208

**Published:** 2020-02-22

**Authors:** Sebastian Scheurer, Salvatore Tedesco, Kenneth N. Brown, Brendan O’Flynn

**Affiliations:** 1Insight Centre for Data Analytics, School of Computer Science and Information Technology, University College Cork, T12 XF62 Cork, Ireland; k.brown@cs.ucc.ie (K.N.B.); brendan.oflynn@tyndall.ie (B.O.); 2Tyndall National Institute, University College Cork, T12 R5CP Cork, Ireland; salvatore.tedesco@tyndall.ie; 3CONNECT Centre for Future Networks and Communications, Tyndall National Institute, University College Cork, T12 R5CP Cork, Ireland

**Keywords:** human activity recognition, machine learning, wearable sensors, inertial sensors, multi-class classification, hierarchical classification, error-correcting output codes, ensembles of nested dichotomies

## Abstract

Human activity recognition (HAR) has become an increasingly popular application of machine learning across a range of domains. Typically the HAR task that a machine learning algorithm is trained for requires separating multiple activities such as walking, running, sitting, and falling from each other. Despite a large body of work on multi-class HAR, and the well-known fact that the performance on a multi-class problem can be significantly affected by how it is decomposed into a set of binary problems, there has been little research into how the choice of multi-class decomposition method affects the performance of HAR systems. This paper presents the first empirical comparison of multi-class decomposition methods in a HAR context by estimating the performance of five machine learning algorithms when used in their multi-class formulation, with four popular multi-class decomposition methods, five expert hierarchies—nested dichotomies constructed from domain knowledge—or an ensemble of expert hierarchies on a 17-class HAR data-set which consists of features extracted from tri-axial accelerometer and gyroscope signals. We further compare performance on two binary classification problems, each based on the topmost dichotomy of an expert hierarchy. The results show that expert hierarchies can indeed compete with one-vs-all, both on the original multi-class problem and on a more general binary classification problem, such as that induced by an expert hierarchy’s topmost dichotomy. Finally, we show that an ensemble of expert hierarchies performs better than one-vs-all and comparably to one-vs-one, despite being of lower time and space complexity, on the multi-class problem, and outperforms all other multi-class decomposition methods on the two dichotomous problems.

## 1. Introduction

Human activity recognition (HAR) systems have become an increasingly popular application area for machine learning across a range of domains, including the medical (e.g., monitoring ambulatory patients [1]), industrial (e.g., monitoring workers for movements with increased risk of repetitive strain injury [2]), home care and assisted living (e.g., monitoring the elderly for dangerous falls [3]) domains. A particularly popular approach which has proven successful in numerous HAR applications is to extract a set of features from inertial data along a sliding window, and use the resulting matrix—whose rows and columns correspond to windows and features, respectively—as inputs to the machine learning algorithm [4,5]. More often than not the HAR task for which a learning algorithm is trained goes beyond separating two activities, requiring the algorithm to distinguish among many different activities such as lying, sitting, standing, walking, nordic walking, running, rowing, and cycling [6]. To apply machine learning classification algorithms that are designed to  deal with two target classes, such a multi-class problem must first be decomposed into a set of binary problems. Then, a separate instance of the machine learning algorithm is trained for each of these binary problems. When a new sample is presented to the system, it is passed to each of the trained classifiers and their outputs, which may be probabilities, are combined [7]. Besides making the multi-class problem amenable to binary classification algorithms, there are other benefits to decomposing a  multi-class problem into a set of dichotomous problems. Perhaps the most important benefit is that a binary classification algorithm can be subjected to any of a number of well-known analyses, such as  Receiver Operator Characteristic (ROC) or Sensitivity/Specificity analysis, which can yield insights that serve to tune the classifier with respect to the relative cost of false positives and false negatives.

There are several methods for transforming a multi-class classification problem into a set of binary classification problems [7,8]. The most popular of these are undoubtedly one-vs-all and, to a lesser extent, one-vs-one. Another approach is based on error-correcting output codes, which may be  learned from labelled or unlabelled data. Finally, there are hierarchical methods in which the classes are arranged in a tree or (in rare cases) in a directed acyclic graph, which may be constructed randomly, learned from the data, or constructed from common sense or  domain knowledge. Such a hierarchical approach, which is often referred to as a *top-down* approach, is particularly appealing in application areas where the concepts (or classes) of interest are naturally arranged in a hierarchy. There are examples of more or less formal class hierarchies in many application domains—such as gene and protein function ontologies, music (and other artistic) genres, and library classification systems—and this has inspired many authors to develop hierarchical classifiers that excel at text categorisation, protein function prediction, music genre classification, and emotional speech and phoneme classification.

In HAR applications, it is almost always natural and easy to arrange the activities of interest in a hierarchy, for example by placing the most general categories (e.g., “mobile” and “stationary”) at the top or root of the tree, proceeding to increasingly specific categories (“walk” and “run”), and terminating with the most specific categories (“walk upstairs” and “walk downstairs”) at the leaves. Furthermore, it is not uncommon that a HAR system’s end users find it difficult to precisely specify which activities need to be recognised, not to mention class priors and misclassification costs, which are needed to properly design data acquisition protocols and tune classifiers. In such situations, nested dichotomies can be useful because they make it possible to develop increasingly fine-grained HAR capabilities iteratively. Having a classifier that can accurately distinguish between, for example, stationary and mobile behaviours at an early stage of the development life cycle not only enables early systems-level testing and end user feedback, but can speed up the annotation process—a task which is error-prone and often requires a disproportionate amount of human effort—for more specific activities. These advantages have inspired several applications of the principle  [1,9,10,11]. Unfortunately, there has been little research into how hierarchical approaches to HAR inference compare to other multi-class decomposition methods, such as one-vs-one and one-vs-all. This is  particularly striking because HAR problems tend to be multi-class problems, and because the performance of classification algorithms can be significantly affected by whether and how the multi-class problem is decomposed into a set of binary classification problems. Thus, it is unclear whether or not the benefits of a hierarchical approach for HAR come at the cost of worse predictive performance, and if so, just how high that cost might be. The main questions addressed in this work are:Does the effect of the multi-class decomposition method choice on a HAR problem reflect what has been reported by other comparative studies of multi-class decomposition methods, namely that one-vs-one tends to perform slightly better than one-vs-all in most, but not all, cases?How does the performance of expert hierarchies compare to that of one-vs-all, which is the de-facto standard in practice? How much does a multi-class classifier stand to gain or lose from the domain knowledge encoded in an expert hierarchy?How does the performance of an ensemble of expert hierarchies compare to that of an equally sized ensemble of random nested dichotomies (i.e., an Ensemble of Nested Dichotomies), and to that  achieved by individual expert hierarchies? The former comparison indicates whether the domain knowledge encoded in the expert hierarchies is useful information or detrimental bias for a  classifier, and the latter whether, given some set of candidate expert hierarchies for a multi-class problem, we should look for and use a single expert hierarchy, or combine them into an  ensemble of expert hierarchies.How do these methods perform when evaluated at an expert hierarchy’s topmost dichotomy, for example to separate “Mobile” from “Stationary”, or “Emergency” from “Not Emergency” activities?

In answering these questions, we make four main contributions: (1) the first empirical evaluation of the effect that the choice of multi-class decomposition method has on the performance of various binary learning algorithms on a multi-class HAR problem; (2) the first direct comparison of hierarchical classification that is guided by domain knowledge with standard domain-agnostic multi-class decomposition methods on a multi-class HAR problem; (3) we formulate a threshold that indicates when a nested dichotomy’s branch cannot possibly be on the path to the predicted class, and therefore does not need to be explored; and (4) we show that domain knowledge can be used to construct a multi-class classifier that has lower computational complexity and is easier to interpret than, but performs comparable to one-vs-all.

The remainder of this paper is organised as follows. The following section describes the most popular multi-class decomposition methods, and the one after that (Section 3) briefly reviews the literature investigating multi-class decomposition methods in a HAR context. Then, Section 4 describes the data and computational experiments, whose results show that expert hierarchies are able to compete with one-vs-all, and indeed many of the other multi-class decomposition methods discussed in Section 2, regardless of whether we look at the results for the original multi-class problem or those for the binary classification problem induced by an expert hierarchy’s topmost dichotomy. The results, presented in Section 5, also show that Ensembles of Expert Hierarchies perform comparably to an equally sized Ensemble of Nested Dichotomies on the multi-class problem, but with  significantly lower variance among both the cross-validation folds and the learning algorithms than the ensemble of nested dichotomies. Section 6 concludes our presentation of the results by summarising and discussing the main findings. Finally, Section 7 concludes the paper.

## 2. Multi-Class Decomposition Methods

This section discusses the multi-class decomposition methods used in this work, which are presented in three groups—flat decomposition strategies (Section 2.1), strategies based on  error-correcting output codes (Section 2.2), and hierarchical strategies (Section 2.3).

### 2.1. Flat Decomposition Strategies

An intuitive approach for decomposing a multi-class problem into a set of binary classification problems is to use an indicator matrix with one column per class that encodes whether or not an observation belongs to that class. This method is known as one-versus-rest, or one-versus-all, and discussed in more detail by Park ([8], p. 16). One-vs-all requires fitting, storing, evaluating, and averaging *k* models for a *k*-class problem, one model per class. It is the default method for handling multi-class classification problems in most machine learning libraries and packages, including Weka [12] and Scikit-learn [13]. A somewhat more elaborate method has become known as pairwise classification, one-versus-other, or one-versus-one [14,15]. One-vs-one (OVO) fits one model for each pair of classes, using only those observations that belong to either of the two classes. One-vs-one requires fitting, storing, and evaluating k(k−1)/2 models, which might explain why, while an implementation is available in most machine learning libraries, it is not the default multi-class decomposition method in any of them. Weka, for example, implements one-vs-one as an option to its MultiClassClassifier class, which  also implements error-correcting and one-vs-all, with the latter being its default multi-class decomposition method [16], and Scikit-learn has a  OneVsOneClassifier which can be used with any classifier conforming to the Scikit-learn API [17] instead of one-vs-all—which is Scikit-learn’s default multi-class decomposition method, too. Class-wise confidence scores for one-vs-one can be calculated by adding the number of votes and the  normalised sum of pairwise confidence levels predicted by the binary classifiers.

### 2.2. Error-Correcting Output-Codes

The idea to use error-correcting output codes (ECOC) for decomposing a multi-class problem was introduced in 1995 by Dietterich and Bakiri [18] who took an information-theoretic perspective and framed the problem as a coding problem. To use the error-correcting output codes approach, one first defines a binary code matrix W, in which each class is represented by one row which contains the code word for that class. Then, a classifier is trained for each column in the code matrix, but with the outcome replaced by the code matrix’s corresponding entry, that is, when fitting classifier *j* we replace each occurrence of class *i* with the entry found in row *i* and column *j* of the binary code matrix W. To recover the *n* classes we apply the *k* classifiers, multiply the output (i.e., probability estimate) from classifier *j* with column vector *j*, and arrange the products in the same order in a matrix $\hat{W}$. Finally, an observation is labelled as belonging to the class whose predicted code (i.e., row in $\hat{W}$) is closest to the corresponding code (row) in the code matrix W, according to some distance metric. The distance function proposed by Dietterich and Bakiri [18] is the $L_{1}$ distance
(1)$$D\left(w_{i},{\hat{w}}_{i}\right)=\sum_{j}\left|{\hat{w}}_{i,j}-w_{i,j}\right|,$$
where *i* iterates over the rows (i.e., classes) and *j* over the columns (i.e., binary classifiers) of the code matrix W. A confidence score for class *i* can be calculated by evaluating $D(w_{i},1-{\hat{w}}_{i})$, that is, by calculating the *L*${}_{1}$ distance between the code matrix and the vector of probabilities predicted by the binary classifiers for each’s respective *negative* class.

Allwein et al. [19] subsequently extended this work—and the design space for the code matrix W—by allowing the entries of W to take on one of three (instead of two) values, namely −1, 0, or +1, where a zero indicates that instances of this class be excluded from the corresponding model, while +1 and −1 encode whether the corresponding code bit is on or off, respectively. This extension makes it possible to encode any possible decomposition, including nested dichotomies, in the code matrix, but does not provide any guidance on how to design a good code matrix. The error-correcting output codes approach has since been taken further in various papers that focus on designing a problem-dependent code matrix for a given multi-class classification problem based on training data. Pujol et al. [20], for example, proposed *Discriminant ECOC* in 2006, which uses floating search to find a nested dichotomy (binary tree) that maximises the quadratic mutual information, which is then represented as a coding matrix of size k−1. More recently, Bautista et al. [21] proposed two evolutionary algorithms, based on *genetic algorithms* and *population based incremental learning*, to find a minimal coding matrix—that is, one with $\left\lceil \log_{2}k\right\rceil$ columns for a *k*-class problem—that achieves good generalisation for a given machine learning algorithm and classification problem.

### 2.3. Hierarchical Decomposition Strategies—Nested Dichotomies

In a nested dichotomy the *k* classes are placed as the *k* leaf nodes of a binary tree. Nested dichotomies are a well-known technique for dealing with a polychotomous response in regression analysis, whose results depend on the particular nested dichotomy used [22], and which are applicable if there is enough domain knowledge to construct an appropriate and justifiable nested dichotomy for a given problem. To construct a nested dichotomy from domain knowledge (or common sense), the *k* classes are placed as the leaf nodes of a binary tree according to a hierarchy of the *k* concepts that represents the domain knowledge. To distinguish a nested dichotomy constructed from domain knowledge in this manner from one constructed by some other method, we refer to the former as an expert hierarchy and to the latter simply as a nested dichotomy. To train a nested dichotomy, an instance of the binary classifier is trained for each internal node of the tree using only the data belonging to either of the classes represented by that node’s children. At prediction time, each of the trained binary classifiers is applied and the outputs aggregated. Because the dichotomies that constitute a nested dichotomy are mutually independent [22], the expected probability that a new instance belongs to a particular class is given by the product of the estimated probabilities that are on the path to the leaf representing that class.

Nested dichotomies have multiple advantages over non-hierarchical multi-class decomposition methods: lower time and space complexity at both training and evaluation (prediction) time, easier interpretation, and a modular architecture that fosters division of labour and iterative development. Time and space complexity at training time is lower for nested dichotomies than for one-vs-all (and, by extension, than for one-vs-one), because fewer binary classifiers need to be fitted, and because each classifier, bar the one at the root of the hierarchy, is only fitted to a subset of the training data. Time and space complexity at evaluation (prediction) time is lower for nested dichotomies, because there are fewer binary classifiers to begin with, and we may not have to evaluate all of them to predict the most likely class label.

As we shall see in Section 3, many authors use a variation of nested dichotomies that might be called a non-probabilistic nested dichotomy. The probabilistic nested dichotomies we use predict the branch probabilities at each internal node, recursively multiplying them with those predicted by its children until arriving at the leaves. A non-probabilistic nested dichotomy, on the other hand, predicts a discrete class at each internal node, only descending into the branch that corresponds to the predicted class and terminating at a single predicted activity label. Both our statistical intuition and the literature suggest that a probabilistic nested dichotomy is preferable to a non-probabilistic nested dichotomy, but non-probabilistic nested dichotomies do have one advantage. Namely, we do not need to apply all its constituent binary classifiers to predict a discrete class label, but can achieve the same outcome with $\log_{2}k$ to k−1 classifiers, depending on whether the hierarchy is balanced or a chain, respectively. However, this aspect of probabilistic nested dichotomies can be improved if we avoid descending into any branch whose predicted probability is too small to compete with the probabilities predicted for its sibling, or any of its sibling’s descendants. This probability threshold depends on the (maximum) depth of the tree below the more likely of the two branches, and on the threshold for converting the predicted probabilities into discrete class predictions. The relationship can be formulated as follows in terms of the more likely branch’s predicted probability
(2)$$p_{y}\geq \frac{1}{t^{d}+1},$$
where $p_{y}$ denotes the predicted probability of the more likely branch (denoted by *y*), *t* the probability threshold (assumed to satisfy 0<t<1), and *d* the depth of the tree attached to node *y*, with d=0 if *y* is itself a leaf node. We can apply Equation (Equation 2) at each internal node and not descend into the less likely of its branches if the more likely branch’s predicted probability $p_{y}$ (which must, by definition, meet the threshold *t*) satisfies Equation (Equation 2), in which case it is certain that the leaf with the largest predicted probability will turn out to be *y* (if it is a leaf node), or, if *y* is an internal (classifier) node, among its descendants.

A binary tree with *k* leaves has k−1 internal (non-leaf) nodes, and hence a nested dichotomy for a *k*-class problem requires fitting and storing k−1 binary classifiers, and evaluating between $\log_{2}k$ and k−1 of them, depending on how often the probability predicted by an internal node’s binary classifier satisfies Equation (Equation 2). The number of all the possible full binary rooted trees with n+1 leaves is given by the *n*-th Catalan number [23]
$$C_{n}=\frac{(2n)!}{(n+1)!n!}.$$

To construct all the possible nested dichotomies for a *k*-class problem would thus require to fit, store, apply, and aggregate the outputs of $\left(k-1\right)C_{k-1}$ models. Because of the rapid growth of this function—for k=4 we have $3C_{3}=18$, for k=7 it is $6C_{6}=792$, and for k=13 we have $12C_{12}=2,496,144$—considering all possible binary trees is intractable for larger values of *k*. Even so, if we have a sound and thorough theoretical understanding of the data generating process, or enough domain knowledge to construct a plausible nested dichotomy (or set of nested dichotomies) for a given problem, then nested dichotomies are a realistic option.

However, we often apply machine learning techniques to problems for which we do not have enough domain knowledge to construct an appropriate nested dichotomy. To overcome this obstacle with nested dichotomies Frank and Kramer [24] introduced the “Ensemble of Nested Dichotomies” in 2004, a technique that was further refined by Dong et al. [25] and Rodríguez et al. [26] in 2005 and 2010, respectively. To construct an ensemble of nested dichotomies for a problem with *k* classes, one draws a random sample (with replacement) of predetermined size *m* from the space of all possible binary nested dichotomies with *k* leaf nodes. Each of these is then separately fitted to the data, resulting in a set of *m* nested dichotomies which are combined into an ensemble classifier by averaging the outputs of the individual nested dichotomies. Because an ensemble of nested dichotomies with *m* members is simply a combination of *m* nested dichotomies, it requires fitting, storing, and evaluating m(k−1) fitted classifiers for a problem with *k* classes.

## 3. Related Works

Given the recent breakthroughs achieved by deep learning in many machine learning application areas—most notably computer vision and natural language processing—and the fact that deep learning models are inherently multi-class, we begin our survey of the literature with a brief summary of deep learning for human activity recognition. We then turn our attention to the literature on multi-class decomposition methods and the impact they have on the performance of classification algorithms.

In 2011, Wang et al. [27] they surveyed 56 papers that use deep learning models—deep neural, convolutional, and recurrent neural networks, autoencoders, and restricted Boltzmann machines—to perform sensor-based human activity recognition. They concluded that there is no single “model that outperforms all others in all situations,” and recommend to choose a model based on the application scenario. They identify four papers [28,29,30,31] as the state of the art in deep learning for HAR, based on a comparison of three HAR benchmark data-sets, viz. the Opportunity [32], Skoda [33], and the UCI (University of California, Irvine) smartphone [34] data-sets, all of which consist of data acquired from subjects wearing multiple inertial measurement units (IMUs). What follows is a summary of these results.

Jiang and Yin [28] proposed DCNN+, a deep convolutional neural network (DCNN) model to recognise human activities from signal and activity images constructed by applying 2D Wavelets or the Discrete Fourier Transform to signals from a single IMU. To improve DCNN performance, they use binary support vector machine (SVM) classifiers to discriminate between pairs of classes whose predicted probabilities are similarly large. Their DCNN+ (DCNN + disambiguating SVMs) achieves the same, or only marginally (0.55 to 1.19 percentage points) higher, accuracy than an SVM operating on the same 561 features used for training the binary SVMs that are used to disambiguate between potentially confused predictions in the DCNN+. On the FUSION data-set [35], the DCNN and DCNN+ approach both achieved the same performance (99.3% accuracy) as the SVM. Zhang et al. [29] proposed a DNN that recognises human activities from the raw signals acquired by a single IMU, and the signal magnitude of the accelerometer’s combined three axes. They compare their method with traditional machine learning algorithms operating on five features (mean, standard deviation, energy, spectral entropy, and pairwise correlations between the accelerometer axes) without tuning any of the algorithms’ hyper-parameters. DNN achieved an error rate of 17.7% (SVM: 19.3%) on the Opportunity data-set, 8.3% (SVM: 22.2%) on USC-HAD [36], and 9.4% (kNN: 22.7%) on the Daily and Sports Activities data-set [37].

Ordóñez and Roggen [30] proposed a Deep Convolutional Long Short-Term Memory cell (LSTM) model. Their proposed method outperformed a baseline Convolutional Neural Network (CNN)—which in turn achieved better performance than the best traditional learning algorithms—by 1.8 percentage points (F-score: 93% vs. 91.2%) on the Opportunity data-set. On the Skoda data-set—which also consists of data from multiple IMUs per subject—their deep convolutional LSTM outperformed the state of the art by 6.5 percentage points (95.8% vs. 89.3%). Hammerla et al. [31] explored the application of DNNs, CNNs, and three different flavours of LSTMs on three benchmark HAR data-sets (Opportunity, PAMAP2 [38], and Daphnet [39]), all of which consist of data from subjects instrumented with multiple IMUs. They explored the impact of various hyper-parameters, which determine the architecture, learning, and regularisation of the various deep models by running hundreds or thousands of experiments with randomly sampled parameter configurations. They, too, found that no model dominates the others across all three data-sets. A bi-directional LSTM achieved the best F-score (92.7%) on the Opportunity data (4% better than the deep convolutional LSTM by Ordóñez and Roggen [30]), a CNN the best score (93.7%) on PAMAP2, and a forward LSTM the best score (76%) on the Daphnet data-set. They further show that tuning the hyper-parameters is critical to achieve good performance, as the best model’s median score was 17.2 percentage points lower than its best score on the Opportunity data, and 7.1 percentage points lower than its best on the PAMAP2 data. The latter quantity represents the smallest discrepancy they found across all models and data-sets. The largest discrepancy was a 29.7 percentage points difference on the Daphnet data.

We recently applied the classification algorithms and features used in this paper to a single IMU for various benchmark data-sets [40]. The results give an idea of how the deep learning results discussed above compare to our approach. Although we presented our results in terms of Cohen’s κ, we have calculated the F-scores, accuracy, and error rate corresponding to the published results. The ensemble of gradient boosted trees also used in this paper achieved an F-score of 88.5% (± 1.6) and an error rate of 10.9% (± 1.5) on the Opportunity data, an accuracy of 98.4% (± 0.3) on the FUSION data, and an F-score of 89.7% (± 0.5%) on the PAMAP2 data-set. These results show that deep learning outperforms traditional machine learning with handpicked features on data from multiple IMUs by a margin of >6%. However, we cannot draw the same conclusion when it comes to HAR with a single IMU—which is more convenient for end users who have to remember to wear and charge the IMUs. Here, deep learning performs comparable, or only marginally better, than traditional machine learning with handpicked features. Furthermore, many papers that demonstrate deep learning methods that outperform traditional machine learning by a large margin compare a deep architecture that was carefully tailored to the data-set and whose hyper-parameters were finely tuned, against machine learning algorithms with default hyper-parameters operating on a handful of basic features. It is, therefore, too early to altogether abandon research into machine learning with handpicked features for HAR.

We now turn to the literature about how multi-class decomposition methods affect the performance of classification algorithms. This discussion is presented in two parts. In the first, we focus on the more popular flat multi-class decomposition methods, such as one-vs-all and one-vs-one, and on multi-class decomposition methods based on error-correcting output codes. The second discusses hierarchical multi-class decomposition methods, such as nested dichotomies and ensembles of nested dichotomies.

Joseph et al. [41], who combined one-vs-one and one-vs-all with a latent variable model, and compared the performance on two DNA micro-array data tumour classification problems [42,43], found that while one-vs-one performed quite clearly better than one-vs-all on one problem (by over 10 percentage points on average), one-vs-all tended to perform better than one-vs-one on the other, albeit only marginally. In 2011, Galar et al. [7], they presented an empirical comparison of one-vs-one and one-vs-all, in which they combined one-vs-one and one-vs-all with SVM, decision trees, *k*-Nearest Neighbours (kNN), Ripper [44], and a positive definite fuzzy classifier [45], and evaluated their performance on 19 publicly available multi-class data-sets. They found that one-vs-one outperformed one-vs-all in almost all cases, although rarely by more than one standard error. Raziff et al. [46] compared one-vs-one, one-vs-all, and error-correcting output codes (with random code matrices of varying size) in combination with decision trees to identify (k=30) people from accelerometer data acquired via a handheld mobile phone, and found that one-vs-one, which achieved an accuracy of 88%, performed better than either one-vs-all or error-correcting output codes, which achieved 70% and 86%, respectively. They also found that when the width of the error-correcting output codes code matrix was increased from *k* to 2k, the accuracy increased by 11%. However, when the width was increased beyond that—to 3k, 4k, and finally 5k—the rate of improvement slowed down to 2% to 3%. These studies show that while one-vs-one is likely to perform better than one-vs-all in most cases, it is not guaranteed to do so for any particular problem.

Hierarchical models in the form of nested dichotomies (a binary hierarchy or tree of binary classifiers) have long been a popular statistical tool for analysing polychotomous response variables [22], where they are usually combined with the binomial logistic regression model to draw inferences about the relationships between predictors and the response. The link between the statistical theory of nested dichotomies (namely that the constituent nested dichotomies are independent) and hierarchical classification in a machine learning context was established when Frank and Kramer [24] introduced the ensemble of nested dichotomies in 2004, and compared its performance to one-vs-one, error-correcting output codes, and one-vs-all on 21 publicly available data-sets. Besides confirming that one-vs-one tends to perform better than one-vs-all, they also found that ensembles of nested dichotomies were comparable to error-correcting output codes and more accurate than one-vs-one when combined with decision trees, and comparable to one-vs-one and more accurate than error-correcting output codes when combined with Logistic Regression, Zimek et al. [47] compared the performance of expert hierarchies with that of ensembles of nested dichotomies—some of which were constrained by an expert hierarchy built from a machine-readable ontology—and a non-binary expert hierarchy with an ensemble of nested dichotomies at internal nodes (HEND). They found that while expert hierarchies improved the performance on simulated data, the HEND performed better on the data-set of real protein expressions. This shows that hierarchical multi-class decomposition methods that are based on domain knowledge can achieve better performance than ensembles of random nested dichotomies on real-world data. Due to the ease of constructing an intuitive hierarchy of increasingly detailed human activities, hierarchical classification has been exploited for HAR by Mathie et al. [9] and Karantonis et al. [1]. Both papers develop a hierarchical classifier for a multi-class HAR problem that is similar to a nested dichotomy. Their hierarchical classifier predicts a discrete activity at internal nodes via hard thresholding, arriving at a single predicted activity label. A nested dichotomy, on the other hand, multiplies the probabilities of the internal nodes on the path to each leaf to predict a probability for each activity, rather than a single activity label.

This non-probabilistic approach—trace the path of discrete “yes” or “no” predictions down the tree until hitting at a leaf, and return its class as the predicted class label—appears to be the norm in the hierarchical classification literature. None of the 74 papers—38 on text categorisation, 25 on protein function prediction, six on music genre classification, three on image classification, and one each on phoneme and emotional speech classification—reviewed by Silla and Freitas [48] in their 2011 survey of hierarchical classification used probabilistic hierarchies, opting instead for non-probabilistic hierarchies that discard their constituent classifiers’ confidence in their predictions. Nevertheless, Silla and Freitas [48] found that hierarchical classification is a better approach to hierarchical classification problems than flat approaches, including not only one-vs-one and one-vs-all, but also inherent multi-class algorithms. More recently, in 2018, Silva-Palacios et al. [49], experimenting with learned, rather than pre-defined, hierarchies across 15 multi-class benchmark data-sets (none of them HAR data) from the UCI machine learning repository [50], reported that probabilistic nested dichotomies clearly tend to outperform, albeit only by a small margin, their non-probabilistic counterparts.

Unfortunately neither of these, nor any of the other comparative studies of multi-class decomposition methods in the literature included a HAR problem in their evaluation, and, because there appears to be no multi-class decomposition method that is dominant across all multi-class classification problems, we cannot assume that one-vs-one, which tends to perform best in most domains, is going to do so in the HAR domain. Furthermore, it can be argued that the activities (concepts), which HAR algorithms are trained to recognise, have a much stronger hierarchical structure than the concepts targeted by most multi-class classification benchmarks, which may affect multi-class decomposition method performance. Moreover, none of the papers that do address the multi-class decomposition problem in a HAR context compares the performance of the proposed method to that of other multi-class decomposition methods such as one-vs-all or one-vs-one. Given the intuitiveness and popularity of hierarchical multi-class decomposition methods for HAR, and their inherent modularity and flexibility, it is important to study whether or not there is a trade-off between using a hierarchical multi-class decomposition method such as an expert hierarchy and using domain-agnostic multi-class decomposition methods such as one-vs-one and one-vs-all, and, if this is the case, estimate how much we stand to gain (or lose) from using a hierarchical multi-class decomposition method that encodes HAR domain knowledge.

## 4. Materials and Methods

### 4.1. Data Description

The data used in our experiments were obtained from a wearable inertial measurement unit (IMU) which is a component of the indoor localisation [51,52] and status monitoring system for emergency first responders developed by the SAFESENS project [53]. These IMUs are equipped with a high-performance low-power 168 MHz 32-bit microprocessor with 1MB of flash memory and 192 KB + 4 KB of random access memory (RAM), a bluetooth low-energy (BLE) communication module, a rechargeable battery, sensors for barometric pressure, humidity, temperature (internal and external), and a tri-axial accelerometer, gyroscope and magnetometer. The inertial sensors connect to the micro-controller over the Inter-Integrated Circuit (I2C) bus, while the environmental sensor adopts the Serial Peripheral Interface (SPI) bus. The board measures 44 mm × 30 mm × 8 mm without battery. Data acquired by the board can be transmitted wirelessly (via BLE), or logged to a removable Micro SD card. To obtain the experimental data-set we recruited 11 volunteers who wore a backpack with an IMU attached to one of the backpack’s straps while performing several trials of each of the 17 activities of interest. The activities, which were chosen to represent activities that are of interest for monitoring emergency first responders during an operation, are given in Table 1 which also lists the proportion with which they are represented in the final data-set. Scheurer et al. [54] have shown that the sensors most useful for human activity recognition are the accelerometer and gyroscope, which are the sensors used for the work at hand, too. To prepare the sensor data for input into the machine learning algorithms, we used the following procedure (which is explained in more detail in [55]). First, the raw signals are smoothed using a median filter with a window size of 3 samples, before being resampled to the mean sampling frequency to obtain regularly sampled signals. The smoothed and resampled accelerometer signal is then separated into its gravity and body components via a low-pass filter as described by Karantonis et al. [1]. Finally, a set of time- and frequency-domain features—namely the mean, standard deviation, skew, kurtosis, inter-quartile range, spectral entropy, peak-power frequency, pairwise correlations between each sensor’s three axes, and the accelerometer’s signal magnitude area—are extracted along a sliding window (with size and overlap of 3 s and 1 s, respectively). These features, together with the ground truth—the activity that the person wearing the IMU was engaged in during the 3 s window summarised by the features—are the inputs used to train and evaluate the machine learning algorithms. The final data-set consists of 8919 instances ×70 features. More details about the experiments, data acquisition protocol, and the data themselves can be found in [55].

### 4.2. Computational Experiments and Evaluation

We estimate the predictive performance of five well-known multi-class decomposition methods (one-vs-all, one-vs-one, ensembles of nested dichotomies, and error-correcting output codes), five expert hierarchies, and an ensemble of expert hierarchies across five machine learning algorithms by means of stratified ten-fold cross-validation. In addition to the three algorithms described and tuned for this particular problem by Scheurer et al. [55]—namely gradient-boosted ensembles of decision trees, binary SVMs, and kNN—we also investigate logistic regression, and decision trees. We further estimate algorithms’ performance when used in their multi-class formulation. Multi-class kernel SVMs have been formulated [56,57,58], but their performance tends to be similar to that of binary SVMs with multi-class decomposition. Furthermore, fitting (and applying) a single non-linear multi-class SVM to a *k*-class problem tends to incur worse computational costs than fitting and applying either *k* binary SVMs with one-vs-all or k(k−1)/2 binary SVMs with one-vs-one [59]. We tried fitting a multi-class SVM with a polynomial kernel using the implementations by Crammer and Singer [57], and Joachims et al. [58], but both algorithms timed out after 24 h without even converging for a single cross validation fold. We therefore use the linear multi-class SVM formulation proposed by Crammer and Singer [57] with default hyper-parameters (C = 1.0 and ϵ = 1 × 10${}^{-4}$).

Prior to passing the data to the machine learning algorithm, each feature is standardised by subtracting its mean and dividing by its standard deviation, both of which are estimated from the cross-validation fold’s training data. We use a random error-correcting output code matrix with 2k=34 columns, which requires about twice as many classifiers as one-vs-all or an expert hierarchy, which require k=17 and k−1 classifiers, respectively, and four times as many as one-vs-one. Five expert hierarchies, which are also used to form an ensemble of expert hierarchies, are constructed by arranging the 17 activities in the data-set as illustrated in Figure A1, Figure A2, Figure A3, Figure A4 and Figure A5 in the Appendix A. To make a fair comparison between ensembles of expert hierarchies and ensembles of nested dichotomies, we construct an ensemble of nested dichotomies with the same number of members as the ensemble of expert hierarchies, namely five. Each expert hierarchy was constructed based on either an engineer’s or an (imaginary) user’s intuition. The engineer’s intuition is to split classes such that the splits are easy to learn for an algorithm, for example because they result in similar patterns in the data. This perspective is represented by EH1 and EH3 (Figure A1 and Figure A3). A user’s intuition, on the other hand, is to split classes such that the earlier splits, which are higher up in the hierarchy, are more informative to them than later splits that are further towards the hierarchy’s leaves. This perspective is represented by EH2 (Figure A2), which considers fall detection, EH4 (Figure A4), which considers separating potential emergencies (in an emergency first response context) from normal behaviours, and EH5 (Figure A5), which considers detecting when someone ascends or descends the stairs.

We further compare multi-class decomposition method (and multi-class) performance on the two binary classification problems corresponding to the topmost (root) dichotomy of EH1 (an example of an engineer’s expert hierarchy) and EH4 (an example of a user’s expert hierarchy). The former separates “Stationary” from “Mobile” and the latter “Possible Emergency” from “Not Emergency” activities. Incidentally, these two splits also provide examples of different levels of class imbalance, with the EH1 split leading to a moderately imbalanced (67%/33%) and the EH4 split to a seriously imbalanced (89%/11%) data-set. The confidence scores obtained with multi-class decomposition methods based on nested dichotomies such as ensembles of nested dichotomies, expert hierarchies, and ensembles of expert hierarchies are true multi-class probabilities (as far as the binary classifier is able to estimate them), and the confidence scores obtained with one-vs-all can easily be combined into multi-class probabilities, but the confidence scores estimated by one-vs-one and error-correcting output codes do not share this characteristic and are prone to be severely affected by class imbalance. To overcome this issue, and give these multi-class decomposition methods a chance to compete on the EH1 and EH4 dichotomies, we calibrate their scores—as well as those estimated by SVM, which is not designed to estimate probabilities even in the binary case—via Platt scaling [60].

The computational experiments were implemented in Python (version 3.7.3), using the sklearn ([13], version 0.20) implementations of machine learning algorithms and multi-class decomposition methods where available (i.e., one-vs-one, one-vs-all, error-correcting output codes, and all machine learning algorithms), and writing our own where necessary, namely for the expert hierarchies, ensembles of expert hierarchies, and ensembles of nested dichotomies. To speed up the experiments they were parallelised using GNU Parallel [61].

## 5. Results

This section presents and analyses the results of the experiments described in Section 4. We use Cohen’s Kappa (κ) statistic as our metric of predictive performance because of its inherent ability to quantify a classifier’s performance on a multi-class classification problem, and because it is adjusted for the prior class distributions of both the ground truth and the predicted class labels.

For a detailed analysis of the differences between the various combinations of machine learning algorithms and multi-class decomposition methods we employ (binomial) logistic regression of the κ statistic on the two factors of interest, viz. the learning algorithm and multi-class decomposition method. The κ statistic, calculated once for each cross-validation test fold, corresponds to the proportion of successful Bernoulli trials—the proportion of test instances classified correctly, adjusted for the probability of chance agreement—and the number of instances in a test fold to the number of trials. Together, these two numbers determine the binomial distribution, allowing us to apply a (binomial) logistic regression model to estimate the log-odds of the κ statistic, $\eta =\ln\frac{\kappa }{1-\kappa }$, which relate to the κ statistic via the logistic function
$$\kappa =g\left(\eta \right)=\frac{e^{\eta }}{e^{\eta }+1}.$$

Because one-vs-all is by far the most popular multi-class decomposition method in practice, and the gradient-boosted ensemble of decision trees (GBT) the algorithm most likely to outperform the others, we use that combination (one-vs-all with GBT) as the baseline (i.e., the regression equation’s intercept) against which the other combinations of multi-class decomposition methods and algorithms are compared. The models were fitted using the R Language and Environment for Statistical Computing ([62], version 3.6.1). In our analysis we limit ourselves to those regression coefficients that are significant at the α = 0.1 significance level.

Table 2 shows the mean κ (in percent, ± its standard error) across the ten cross-validation folds for each multi-class decomposition method. The column labelled “Avg.” lists the mean and standard error (SE) for each multi-class decomposition method, computed across the five machine learning algorithms, and the two rows labelled “Avg.” the mean and standard error over the preceding five rows. Figure 1 illustrates normal (Gaussian) 99% confidence intervals (C.I.) calculated from the means and standard errors given in Table 2. Clearly, the variance between expert hierarchies is negligible compared to that between the other multi-class decomposition methods, and there is no a priori reason to prefer any particular expert hierarchy over the others. Therefore, we pooled the five expert hierarchies (EH1, EH2, …, EH5) into a single category labelled “EH”, and then fitted the regression model to the data summarised in Table 2 to estimate coefficients for seven, rather than eleven, multi-class decomposition methods—one-vs-all (OVA, the baseline/intercept), one-vs-one (OVO), ensembles of nested dichotomies (END), error-correcting output codes (ECOC), multi-class (MCL), expert hierarchies (EH, with no distinction between individual hierarchies), and ensembles of expert hierarchies (EEH)—and five machine learning algorithms, namely ensembles of gradient boosted trees (the baseline/intercept), (binary) SVM, multi-class SVM (SVM-MCL), decision trees (DT), kNN, and logistic regression (GLM).

Table 3 and Table 4 show the mean κ (± standard error), in percent, when evaluating each multi-class decomposition method/machine learning algorithm combination on the topmost (root) dichotomy of EH1 (“Stationary” vs. “Mobile”) and EH4 (“Possible Emergency” vs. “Not Emergency”), respectively. The column labelled “Avg.” lists the mean κ (± standard error), again in percent, across the five machine learning algorithms for each multi-class decomposition method, and the rows labelled “Avg.” the mean and its standard error across the preceding five rows. Figure 2 illustrates the 99% C.I.s for each combination of multi-class decomposition method and learning algorithm based on the means and standard errors in Table 3 and Table 4.

The results of our analysis of the data from the multi-class problem summarised in Table 2 are given in Table 5, and those for the dichotomous problems induced by EH1 and EH4 (Table 3 and Table 4) are given in Table 6 and Table 7, respectively. The tables list the estimate (β) along with its 99% C.I. and *p*-value for those coefficients that are significant at the α = 0.1 level, that is, those with *p* < 0.1. The row labelled “(Intercept)” corresponds to the baseline method’s (OVA ∧ GBT) estimated log odds. For example, the log odds for OVA ∧ GBT on the multi-class problem are estimated as β≈2.99. Therefore, the odds ratio is $e^{\beta }\approx e^{2.99}\approx 19.9$ and hence $\kappa \approx 100\frac{19.9}{19.9+1}\approx 95.2\%$. The other coefficients’ estimates and C.I.s indicate the marginal change in log-odds associated with the corresponding multi-class decomposition method (MDM), learning algorithm, or combination of multi-class decomposition method and learning algorithm. Note that because a positive coefficient signifies an increase in the odds and a negative coefficient a decrease in the odds, a coefficient with a C.I. that spans zero is not significant at the α = 0.01 significance level. Coefficients labelled with a multi-class decomposition method, rather than a combination of multi-class decomposition method and algorithm, estimate the marginal effect that the multi-class decomposition method has on algorithm performance and therefore apply when the multi-class decomposition method is combined with any of the algorithms. Conversely, coefficients labelled with an algorithm, rather than a combination of algorithm and multi-class decomposition method, estimate the marginal effect that the algorithm has on multi-class decomposition method performance, and thus apply when the algorithm is combined with any of the multi-class decomposition methods. Finally, these independent multi-class decomposition method and algorithm coefficients may be amplified or attenuated by a coefficient labelled with a combination of multi-class decomposition method and algorithm (“MDM ∧ algorithm”). These interaction coefficients apply in addition to the independent multi-class decomposition method and algorithm coefficients.

The following examples serve to illustrate these concepts. Consider the logistic regression (GLM) estimates for the multi-class problem from Table 5. The “(Intercept)” (GBT ∧ OVA) is estimated at 2.99, corresponding to odds of $e^{2.99}\approx 19.9$, and hence to a mean κ of $e^{2.99}/\left(e^{2.99}+1\right)\;\approx \;19.9/\left(19.9+1\right)\;\approx \;95.2\%$. An estimate of −1.38 means that the GLM odds are $e^{-1.38}\approx 0.25$ times the baseline odds, that is, $e^{-1.38}e^{2.99}=e^{2.99-1.38}\approx 5.0$ which is equivalent to a mean κ of $e^{2.99-1.38}/\left(e^{2.99-1.38}+1\right)\approx \;83.3\%$. This estimate does not significantly change when GLM is combined with an expert hierarchy or an ensemble of expert hierarchies, as is attested by the absence of the corresponding coefficients from Table 5. However, when GLM is combined with an ensemble of nested dichotomies (END), the estimated odds change by a factor of $e^{-0.15}\approx 0.861$, corresponding to a change of 100(0.861)−100=−13.9% and a mean κ of $e^{2.99-1.38-0.15}/\left(e^{2.99-1.38-0.15}+1\right)\approx 81.2\%$. Note that decision tree (DT) is the only other algorithm whose END performance is significantly different (by a factor of $e^{0.32}\approx 1.377$) from its baseline (one-vs-all) performance. When GLM is applied in its multi-class formulation its odds are subject to the multi-class effect (MCL) that applies to all algorithms, estimated as a $100e^{0.15}-100\approx 16.2\%$ change, which corresponds to a mean κ of $e^{2.99+0.15-1.38}/\left(e^{2.99+0.15-1.38}+1\right)\approx 85.3\%$ for logistic regression. Note that an estimate of −0.15 for the “MCL ∧ kNN” coefficient means that the 16.2% improvement does not hold for kNN, and that an estimate of −0.61 for the “MCL ∧ DT” coefficient, which equates to a $e^{0.15-0.61}\approx -36.9\%$ change in the odds, means that decision trees perform better with one-vs-all than in its multi-class formulation. Finally, let us consider the “ECOC ∧ GLM” combination. When combined with one-vs-all, the log-odds for GLM are 2.99−1.38≈1.61. This baseline estimate is subject to the −0.26 change associated with error-correcting output codes overall, and an additional −0.31 change specific to the “ECOC ∧ GLM” interaction, accumulating in odds that are only $100e^{-0.26-0.31}=e^{-0.57}\approx 56.6\%$, equivalent to a mean κ of $100e^{1.61-0.57}/\left(e^{1.61-0.57}+1\right)\approx 73.9\%$, of logistic regression’s baseline odds.

## 6. Discussion

Our analysis shows that the ensemble of gradient boosted trees significantly and consistently outperforms the other algorithms, both on the original 17-class problem and on the two dichotomous problems induced by the topmost dichotomy of EH1 and EH4. On all three problems, the next best learning algorithm tends to be SVM, followed by decision trees, kNN, and finally logistic regression and the multi-class SVM. While there is no such clear ranking for the multi-class decomposition methods, there are some discernible patterns. Logistic regression, decision trees, and the multi-class SVM are more sensitive to the choice of multi-class decomposition method than the other learning algorithms. Decision trees consistently achieve their best performance when combined with error-correcting output codes. In fact, combining decision trees with error-correcting output codes achieves a κ on the 17-class problem that is only 0.01 percentage points lower than the 95.82% achieved by a multi-class ensemble of gradient boosted trees, our best result on this problem. With any other algorithm, error-correcting output codes perform comparably or worse than one-vs-all, making it one of the worse multi-class decomposition methods for this problem. This is particularly true for logistic regression, which achieves its worst result on all three problems with error-correcting output codes. One-vs-one, which many studies found to perform slightly better than one-vs-all, does not consistently outperform one-vs-all in our evaluation, nor does it achieve the top result for any of our three classification problems. One-vs-one performs significantly (at the α = 0.01 significance level) better than one-vs-all on the 17-class problem when combined with logistic regression or the multi-class SVM, the EH1 dichotomy when combined with decision trees or the multi-class SVM, and on the EH4 dichotomy when combined with decision trees. Furthermore, one-vs-one achieves significantly worse performance on the EH4 problem when combined with SVM, where it achieves 31.6% lower odds than one-vs-all, or kNN, where it achieves 39.3% lower odds than one-vs-all. Applying an algorithm’s multi-class formulation performs significantly (at the α = 0.01 significance level) better than one-vs-all on the topmost EH1 dichotomy when combined with decision trees or logistic regression, and on the topmost EH4 dichotomy when combined with decision trees. Otherwise, an algorithm’s multi-class formulation performs comparably to one-vs-all.

Performance varies much less among expert hierarchies than among the other multi-class decomposition methods, which indicates that any reasonable expert hierarchy is a reasonable choice, and searching for better hierarchies is unlikely to yield significant improvements. Expert hierarchies perform comparably or better than one-vs-all with most algorithms on all three problems. One exception is decision trees, which achieve 24.4% lower odds on the 17-class problem with expert hierarchies than with one-vs-all. The other exceptions are SVM (both in its binary and multi-class formulation), the ensemble of gradient boosted trees, and logistic regression, all of which achieve 13.9% lower odds on the topmost dichotomy of EH4 with expert hierarchies than with one-vs-all. Ensembles of nested dichotomies perform comparably or better than one-vs-all with all but one algorithm. That exception is logistic regression on both the 17-class problem, where it achieves 13.9% lower odds with an ensemble of nested dichotomies than with one-vs-all, and the binary problem induced by the topmost dichotomy of EH4, where it achieves 18.9% lower odds than with one-vs-all. Ensembles of expert hierarchies, on the other hand, perform comparably or better than one-vs-all with all algorithms on all three problems. This makes an ensemble of expert hierarchies a better multi-class decomposition method for this problem than an arbitrary ensemble of (random) nested dichotomies, which may be more difficult to justify to a domain expert.

These results show that expert hierarchies can compete with other multi-class decomposition methods and inherent multi-class classifiers. As mentioned in the introduction, expert hierarchies have two main advantages over both multi-class classifiers and domain agnostic multi-class decomposition methods. The first advantage is iterative and modular development, and the second is targeted tuning and optimisation. Iterative and modular development can speed up and facilitate many of the tasks involved in designing, developing, and maintaining and improving a HAR system. Data annotation is often the most (human) time consuming part of HAR development. With an inherent multi-class classification algorithm, predictive modelling must wait until a data-set has been annotated with all the activities of interest, and be repeated if a new class is introduced. A new class can be introduced if a requirement emerges to distinguish between different types of some higher-level activity. For example, it might be decided upon further consultation with professionals that a HAR system developed for monitoring firefighters’ during operations really ought to distinguish between crawling on one’s hands and knees, and military style on one’s stomach. The distinction is an important one, because smoke tends to rise which makes it important to keep as close to the ground as possible. With one-vs-all it is possible, at least in principle, to begin modelling as soon as the annotations for one class (say, standing) are complete. However, the class imbalance inherent to a one-vs-all decomposition (e.g., “standing” vs. “not standing”) means that any insights gleaned from the modelling will be heavily biased and may not apply to the other dichotomisers. Furthermore, it is probably less efficient and possibly more error-prone to go through a data-set (e.g., fast-forward through hours of video footage) and annotate every time the subject is, or ceases to be, standing, than to annotate when subjects transition between, for example, stationary and mobile behaviour. With expert hierarchies, annotators can generate high-level annotations (e.g., stationary versus mobile) and hand them over to the data science team. The data scientists can then develop and tune the top-level discriminator, knowing that the degree to which they succeed in developing an accurate discriminator for the given labels is directly linked to the system’s overall accuracy. Furthermore, the independence of the dichotomisers that constitute an expert hierarchy makes it possible to replace any of them with a pre-trained model. This means that it is in principle possible to integrate models that have been developed by a third party and fitted to data private or confidential to them, be it to improve the expert hierarchy by replacing an existing dichotomiser or to extend the expert hierarchy with the capacity to make a finer-grained distinction by replacing a leaf in the expert hierarchy with a dichotomiser.

Targeted tuning and optimisation of HAR inference capabilities makes it possible to not only identify problematic activities (e.g., activities with high misclassification costs that tend to be confused with each other), but to effectively improve the performance on the problematic activities without negatively affecting performance on the other activities. Each dichotomiser in an expert hierarchy is an independent binary classifier whose performance can not only be analysed and tuned, but which can can be swapped out for a different algorithm. If the resulting dichotomiser is more accurate than the one it is replacing, then it is bound to improve the multi-class performance. While it is easy to aggregate the probabilities predicted by a true multi-class classifier or some multi-class decomposition method according to an expert hierarchy, we cannot map performance at some internal node of the hierarchy to a single classifier. The independence between an expert hierarchy’s constituent dichotomies makes it easier to explain a prediction to someone without a background in machine learning. Instead of having to simultaneously examine and balance the predicted probabilities of multiple classifiers, none of which says much about the probability distribution over all classes, we can easily identify and examine the output of the binary classifier corresponding to the level at which the prediction first went wrong. Because that classifier is independent of its ancestors and because its own performance has no effect on its descendants’, we can focus our efforts on improving a single binary classifier without having to worry about negatively affecting the performance on other classes.

## 7. Conclusions

We presented the first empirical comparison of the merits of different multi-class decomposition methods for human activity recognition, which covers not only the most popular methods from the literature, namely one-vs-all, one-vs-one, error-correcting output codes, and ensembles of nested dichotomies, but also nested dichotomies that are constructed from domain knowledge, which we call expert hierarchies, and ensembles of expert hierarchies. An expert hierarchy has the advantage that it requires one less binary classifier than one-vs-all, which requires *k* classifiers to represent a *k*-class problem, and that it results in a multi-class decomposition that is easier to interpret than that resulting from one-vs-all. In particular, an expert hierarchy can be designed such that it separates the two most important general concepts—for example “Potential Emergency” and “Not An Emergency”—first, that is, at the topmost level of the hierarchy. With an expert hierarchy it is possible to obtain an estimate for the topmost dichotomy using only a single model (the one corresponding to the topmost dichotomy), which is impossible with any other multi-class decomposition method. We demonstrated this scenario by comparing the predictive performance on the binary classification problem induced by the topmost dichotomy of two example expert hierarchies. Finally, we formulated a threshold that can be used to further reduce the computational complexity of predicting the most likely class label with expert hierarchies—or any nested dichotomy, since an expert hierarchy is just a special case of a nested dichotomy.

The results show that expert hierarchies perform comparably to one-vs-all, both on the original multi-class problem, and on a more general binary classification problem such as that induced by an expert hierarchy’s topmost dichotomy. Our results further show that individual expert hierarchies tend to perform similarly, particularly when compared to the much larger variance among other multi-class decomposition methods or learning algorithms. When multiple expert hierarchies are combined into an ensemble, they perform comparably to one-vs-one and better than one-vs-all on the full multi-class problem, and outperform all multi-class decomposition methods on the two dichotomous problems. Because an expert hierarchy’s constituent dichotomisers are independent of each other it is possible to analyse and optimise each dichotomiser in isolation. This enables modular and iterative development of increasingly complex HAR capabilities, which is a pre-requisite for agile development techniques, and targeted tuning and optimisation of the resulting HAR system.

These results were obtained with a single data-set, and we cannot therefore assume that they will hold up for other HAR problems. They do, however, show that expert hierarchies can have merits in some applications, and justify further research of expert hierarchies. In future work, we therefore plan to evaluate their merits on benchmark HAR data-sets, and investigate whether their potential for integrating data-sets with different activities and transfer HAR models from one set of data and activities to another. 

## Figures and Tables

**Figure 1 sensors-20-01208-f001:**
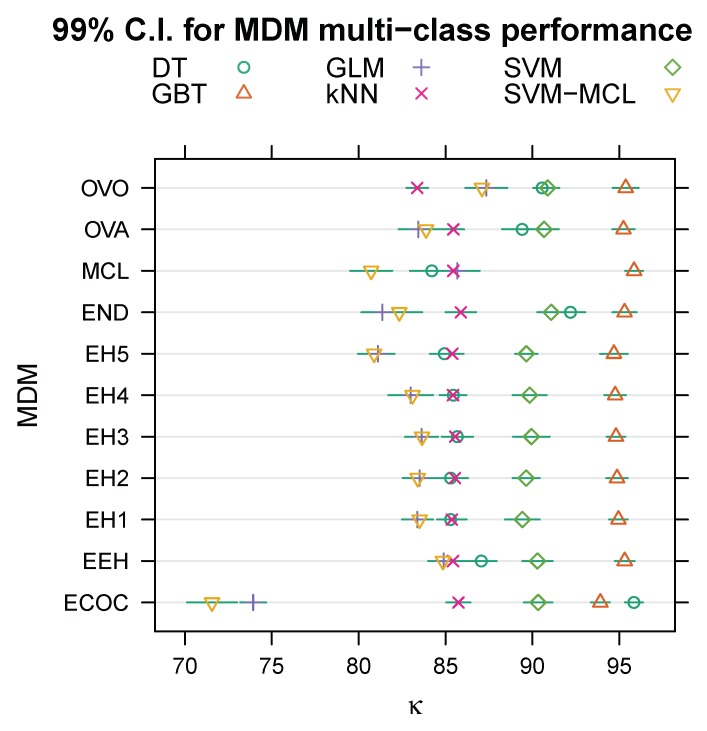
99% confidence intervals (C.I.) for the effect of the multi-class decomposition method (MDM) on the Kappa statistic for the full 17-class problem.

**Figure 2 sensors-20-01208-f002:**
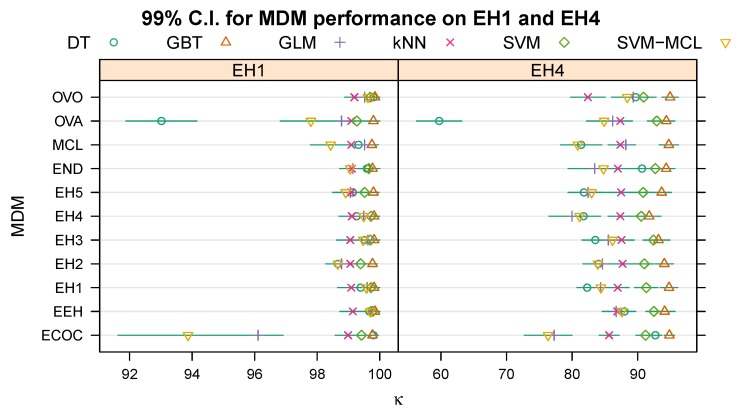
99% confidence intervals (C.I.) for the effect of the multi-class decomposition method (MDM) on the Kappa statistic for the topmost dichotomy of EH1 (left) and EH4 (right).

**Table 1 sensors-20-01208-t001:** The 17 activities and the proportion (%) with which they are represented in the data-set.

Activity	%
All 4 s	5.5
Crouch	4.2
Lie	5.7
Sit	9.3
Stand	8.2
Fall	1.7
Jump Up	1.8
Jump Down	1.8
Crawl Hands & Knees	5.8
Military Crawl	4.8
Duck Walk	3.9
Walk Horizontally	5.7
Walk Down	8.0
Walk Up	9.8
Run Horizontally	12.4
Run Down	5.7
Run Up	5.6

**Table 2 sensors-20-01208-t002:** Mean κ% (± SE) on the multi-class human activity recognition (HAR) problem for each machine learning algorithm and multi-class decomposition method. The first six multi-class decomposition methods are given in order of decreasing average score, the expert hierarchies ordered alphabetically.

	GBT	SVM	DT	kNN	GLM	SVM-MCL	Avg.
OVO	95.37 ± 0.29	90.88 ± 0.26	90.56 ± 0.20	83.37 ± 0.24	87.35 ± 0.47	87.10 ± 0.36	89.11 ± 1.68
END	95.30 ± 0.27	91.08 ± 0.31	92.19 ± 0.33	85.88 ± 0.34	81.37 ± 0.47	82.34 ± 0.51	88.03 ± 2.32
OVA	95.24 ± 0.25	90.67 ± 0.33	89.41 ± 0.45	85.45 ± 0.24	83.43 ± 0.44	83.88 ± 0.53	88.01 ± 1.88
EEH	95.31 ± 0.22	90.29 ± 0.34	87.06 ± 0.34	85.43 ± 0.32	84.89 ± 0.32	84.85 ± 0.33	87.97 ± 1.69
MCL	95.85 ± 0.20	-	84.21 ± 0.49	85.45 ± 0.24	85.67 ± 0.50	80.72 ± 0.47	86.38 ± 2.53
ECOC	93.91 ± 0.21	90.33 ± 0.32	95.84 ± 0.20	85.74 ± 0.27	73.93 ± 0.29	71.56 ± 0.56	85.22 ± 4.20
Avg.	95.16 ± 0.27	90.65 ± 0.15	89.88 ± 1.65	85.22 ± 0.38	82.77 ± 1.95	81.74 ± 2.22	87.45 ± 0.57
EH1	94.95 ± 0.21	89.42 ± 0.39	85.29 ± 0.19	85.36 ± 0.33	83.37 ± 0.34	83.51 ± 0.30	86.98 ± 1.83
EH2	94.87 ± 0.24	89.64 ± 0.30	85.29 ± 0.39	85.54 ± 0.28	83.51 ± 0.37	83.40 ± 0.34	87.04 ± 1.82
EH3	94.81 ± 0.21	89.94 ± 0.41	85.68 ± 0.35	85.55 ± 0.26	83.62 ± 0.37	83.65 ± 0.30	87.21 ± 1.79
EH4	94.76 ± 0.24	89.84 ± 0.38	85.45 ± 0.28	85.43 ± 0.30	82.99 ± 0.50	83.10 ± 0.39	86.93 ± 1.87
EH5	94.69 ± 0.31	89.65 ± 0.25	84.93 ± 0.32	85.39 ± 0.25	81.11 ± 0.37	80.89 ± 0.36	86.11 ± 2.16
Avg.	94.82 ± 0.04	89.70 ± 0.09	85.33 ± 0.12	85.45 ± 0.04	82.92 ± 0.46	82.91 ± 0.51	86.85 ± 0.19

**Table 3 sensors-20-01208-t003:** Mean κ% (± SE) for the topmost dichotomy of expert hierarchy (EH1) (Stationary vs. $\tilde{M}$obile). The first six multi-class decomposition methods are given in order of decreasing average score, the expert hierarchies in alphabetical order.

	GBT	SVM	DT	kNN	GLM	SVM-MCL	Avg.
EEH	99.85 ± 0.06	99.77 ± 0.09	99.67 ± 0.08	99.14 ± 0.16	99.72 ± 0.10	99.70 ± 0.07	99.64 ± 0.10
OVO	99.85 ± 0.06	99.70 ± 0.07	99.80 ± 0.06	99.19 ± 0.12	99.52 ± 0.10	99.62 ± 0.08	99.61 ± 0.10
END	99.77 ± 0.09	99.65 ± 0.10	99.59 ± 0.08	99.11 ± 0.15	99.14 ± 0.11	99.06 ± 0.13	99.39 ± 0.13
MCL	99.75 ± 0.08	-	99.32 ± 0.11	99.09 ± 0.17	99.52 ± 0.14	98.43 ± 0.25	99.22 ± 0.23
ECOC	99.77 ± 0.07	99.42 ± 0.11	99.80 ± 0.06	98.99 ± 0.16	96.11 ± 0.31	93.87 ± 0.87	97.99 ± 1.00
OVA	99.80 ± 0.08	99.27 ± 0.20	93.02 ± 0.44	99.09 ± 0.17	98.78 ± 0.13	97.80 ± 0.38	97.96 ± 1.02
Avg.	99.80 ± 0.02	99.56 ± 0.09	98.53 ± 1.11	99.10 ± 0.03	98.80 ± 0.55	98.08 ± 0.89	98.97 ± 0.32
EH1	99.82 ± 0.07	99.72 ± 0.10	99.39 ± 0.09	99.09 ± 0.17	99.60 ± 0.08	99.57 ± 0.07	99.53 ± 0.11
EH2	99.77 ± 0.06	99.39 ± 0.17	98.66 ± 0.14	99.06 ± 0.15	98.78 ± 0.15	98.66 ± 0.15	99.05 ± 0.18
EH3	99.82 ± 0.07	99.70 ± 0.08	99.52 ± 0.10	99.06 ± 0.17	99.62 ± 0.10	99.47 ± 0.12	99.53 ± 0.11
EH4	99.82 ± 0.07	99.72 ± 0.10	99.26 ± 0.16	99.11 ± 0.16	99.49 ± 0.12	99.49 ± 0.12	99.48 ± 0.11
EH5	99.80 ± 0.07	99.52 ± 0.12	99.14 ± 0.15	99.09 ± 0.17	99.06 ± 0.13	98.91 ± 0.16	99.25 ± 0.14
Avg.	99.81 ± 0.01	99.61 ± 0.07	99.19 ± 0.15	99.08 ± 0.01	99.31 ± 0.17	99.22 ± 0.18	99.37 ± 0.09

**Table 4 sensors-20-01208-t004:** Mean κ% (± SE) for the topmost dichotomy of EH4 (Possible Emergency vs. non-Emergency). The first six multi-class decomposition methods are given in order of decreasing average score, the five expert hierarchies in alphabetical order.

	GBT	SVM	DT	kNN	GLM	SVM-MCL	Avg.
EEH	94.11 ± 0.61	92.46 ± 0.47	87.99 ± 0.66	86.86 ± 0.79	86.71 ± 0.82	87.59 ± 0.70	89.29 ± 1.30
OVO	94.93 ± 0.48	90.90 ± 0.73	89.69 ± 0.63	82.41 ± 1.02	89.32 ± 0.87	88.41 ± 0.93	89.28 ± 1.66
END	94.34 ± 0.51	92.68 ± 0.54	90.66 ± 0.61	86.97 ± 0.89	83.44 ± 1.57	84.79 ± 1.30	88.81 ± 1.80
MCL	94.74 ± 0.57	-	81.38 ± 1.22	87.35 ± 0.72	88.18 ± 0.57	80.87 ± 0.86	86.50 ± 2.54
ECOC	94.83 ± 0.30	91.19 ± 0.57	92.69 ± 0.39	85.65 ± 0.59	77.26 ± 0.90	76.34 ± 1.41	86.33 ± 3.26
OVA	94.36 ± 0.50	92.91 ± 0.56	59.74 ± 1.34	87.35 ± 0.72	86.20 ± 1.12	84.89 ± 1.05	84.24 ± 5.14
Avg.	94.55 ± 0.13	92.03 ± 0.41	83.69 ± 5.04	86.10 ± 0.78	85.19 ± 1.78	83.81 ± 1.84	87.41 ± 0.84
EH1	94.78 ± 0.51	91.30 ± 0.69	82.30 ± 0.61	86.95 ± 0.67	84.35 ± 1.24	84.39 ± 1.08	87.34 ± 1.95
EH2	94.06 ± 0.54	91.00 ± 0.55	84.02 ± 0.70	87.70 ± 0.67	84.62 ± 0.83	83.93 ± 0.88	87.55 ± 1.72
EH3	93.16 ± 0.68	92.39 ± 0.61	83.54 ± 0.79	87.54 ± 0.74	85.52 ± 0.84	86.17 ± 0.69	88.05 ± 1.59
EH4	91.75 ± 0.69	90.53 ± 0.44	81.74 ± 0.59	87.35 ± 0.72	79.98 ± 1.36	81.15 ± 1.23	85.42 ± 2.09
EH5	93.65 ± 0.58	90.82 ± 0.55	81.81 ± 0.94	87.48 ± 0.84	82.43 ± 1.03	82.97 ± 0.83	86.53 ± 2.01
Avg.	93.48 ± 0.51	91.21 ± 0.32	82.68 ± 0.46	87.40 ± 0.13	83.38 ± 0.99	83.72 ± 0.83	86.98 ± 0.46

**Table 5 sensors-20-01208-t005:** Estimated logistic regression coefficients with *p* < 0.1 for the multi-class problem.

Coefficient	0.5%	β	99.5%	*p*
(Intercept)	2.87	2.99	3.13	<2.0 × 10^−32^
SVM	−0.88	−0.72	−0.56	1.0 × 10^−31^
DT	−1.02	−0.86	−0.7	<2.0 × 10^−32^
kNN	−1.38	−1.22	−1.08	<2.0 × 10^−32^
GLM	−1.53	−1.38	−1.23	<2.0 × 10^−32^
SVM-MCL	−1.5	−1.35	−1.2	<2.0 × 10^−32^
ECOC	−0.43	−0.26	−0.09	9.6 × 10^−5^
ECOC ∧ SVM	0.0	0.22	0.44	8.6 × 10^−3^
ECOC ∧ DT	1.03	1.26	1.5	<2.0 × 10^−32^
ECOC ∧ kNN	0.08	0.28	0.49	3.5 × 10^−4^
ECOC ∧ GLM	−0.51	−0.31	−0.12	3.8 × 10^−5^
ECOC ∧ SVM-MCL	−0.66	−0.47	−0.27	9.3 × 10^−10^
EEH ∧ DT	−0.46	0.24	−0.02	4.3 × 10^−3^
EH ∧ DT	−0.46	−0.28	−0.11	1.9 × 10^−5^
END ∧ DT	0.09	0.32	0.55	3.0 × 10^−4^
END ∧ GLM	−0.36	−0.15	0.05	5.5 × 10^−2^
MCL	−0.04	0.15	0.33	4.6 × 10^−2^
MCL ∧ DT	−0.83	−0.61	−0.38	2.4 × 10^−12^
MCL ∧ kNN	−0.36	−0.15	0.07	8.5 × 10^−2^
OVO ∧ kNN	−0.4	−0.19	0.02	2.2 × 10^−2^
OVO ∧ GLM	0.07	0.29	0.5	5.4 × 10^−4^
OVO ∧ SVM-MCL	0.02	0.23	0.44	5.2 × 10^−3^

**Table 6 sensors-20-01208-t006:** Estimated logistic regression coefficients with *p* < 0.1 for the binary problem induced by the topmost dichotomy of EH1.

Coefficient	0.5%	β	99.5%	*p*
(Intercept)	5.65	6.2	6.87	<2.0 × 10${}^{-32}$
SVM	−2.03	−1.29	−0.64	1.2 × 10${}^{-6}$
DT	−4.29	−3.61	−3.05	<2.0 × 10${}^{-32}$
kNN	−2.24	−1.51	−0.88	6.2 × 10${}^{-9}$
GLM	−2.52	−1.8	−1.19	1.3 × 10${}^{-12}$
SVM-MCL	−3.1	−2.4	−1.82	1.6 × 10${}^{-22}$
ECOC ∧ DT	2.71	3.73	4.8	1.9 × 10${}^{-20}$
ECOC ∧ GLM	−1.95	−1.07	−0.17	1.8 × 10${}^{-3}$
ECOC ∧ SVM-MCL	−1.81	−0.95	−0.07	4.6 × 10${}^{-3}$
EEH ∧ SVM	−0.26	0.89	2.03	4.4 × 10${}^{-2}$
EEH ∧ DT	1.77	2.83	3.88	3.0 × 10${}^{-12}$
EEH ∧ GLM	0.09	1.2	2.29	4.7 × 10${}^{-3}$
EEH ∧ SVM-MCL	0.62	1.71	2.78	3.6 × 10${}^{-5}$
EH ∧ SVM	−0.15	0.58	1.39	4.9 × 10${}^{-2}$
EH ∧ DT	1.52	2.17	2.91	4.7 × 10${}^{-16}$
EH ∧ GLM	−0.17	0.52	1.3	6.4 × 10${}^{-2}$
EH ∧ SVM-MCL	0.33	1.0	1.75	2.7 × 10${}^{-4}$
END ∧ SVM	−0.15	0.85	1.87	2.9 × 10${}^{-2}$
END ∧ DT	2.09	3.03	3.99	1.3 × 10${}^{-16}$
END ∧ SVM-MCL	0.08	0.99	1.9	4.8 × 10${}^{-3}$
MCL ∧ DT	1.73	2.61	3.52	2.8 × 10${}^{-14}$
MCL ∧ GLM	0.23	1.16	2.12	1.4 × 10${}^{-3}$
OVO ∧ DT	2.2	3.32	4.45	1.4 × 10${}^{-14}$
OVO ∧ SVM-MCL	0.42	1.49	2.53	2.4 × 10${}^{-4}$

**Table 7 sensors-20-01208-t007:** Estimated logistic regression coefficients with *p* < 0.1 for the binary problem induced by the topmost dichotomy of EH4.

Coefficient	0.5%	β	99.5%	*p*
(Intercept)	2.7	2.82	2.94	<2.0 × 10${}^{-32}$
SVM	−0.4	−0.25	−0.09	7.0 × 10${}^{-5}$
DT	−2.56	−2.42	−2.29	8.9 × 10${}^{-2}$
kNN	−1.03	−0.88	−0.74	<2.0 × 10${}^{-32}$
GLM	−1.13	−0.99	−0.84	<2.0 × 10${}^{-32}$
SVM-MCL	−1.23	−1.09	−0.95	<2.0 × 10${}^{-32}$
ECOC ∧ SVM	−0.55	−0.33	−0.1	1.6 × 10${}^{-4}$
ECOC ∧ DT	1.85	2.05	2.26	<2.0 × 10${}^{-32}$
ECOC ∧ kNN	−0.44	−0.24	−0.03	2.9 × 10${}^{-3}$
ECOC ∧ GLM	−0.9	−0.7	−0.5	1.3 × 10${}^{-19}$
ECOC ∧ SVM-MCL	−0.84	−0.65	−0.45	4.1 × 10${}^{-17}$
EEH ∧ DT	1.45	1.64	1.84	<2.0 × 10${}^{-32}$
EEH ∧ SVM-MCL	0.07	0.27	0.47	4.3 × 10${}^{-4}$
EH	−0.28	−0.15	−0.03	2.0 × 10${}^{-3}$
EH ∧ DT	1.18	1.32	1.47	<2.0 × 10${}^{-32}$
EH ∧ kNN	0.0	0.16	0.32	9.0 × 10${}^{-3}$
END ∧ DT	1.68	1.88	2.08	<2.0 × 10${}^{-32}$
END ∧ GLM	−0.41	−0.21	−0.01	6.0 × 10${}^{-3}$
MCL ∧ DT	0.81	1.01	1.2	<2.0 × 10${}^{-32}$
OVO	−0.06	0.11	0.29	8.9 × 10${}^{-2}$
OVO ∧ SVM	−0.61	−0.38	−0.16	8.9 × 10${}^{-6}$
OVO ∧ DT	1.45	1.66	1.86	<2.0 × 10${}^{-32}$
OVO ∧ kNN	−0.7	−0.5	−0.3	2.2 × 10${}^{-10}$
OVO ∧ GLM	−0.03	0.18	0.39	2.7 × 10${}^{-2}$
OVO ∧ SVM-MCL	−0.01	0.19	0.4	1.6 × 10${}^{-2}$

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
