# Peer review of "Using Domain Knowledge for Interpretable and Competitive Multi-Class Human Activity Recognition"

_sensors, 2020, doi:10.3390/s20041208_

Round 1

Reviewer 1 Report

The manuscript "Using Domain Knowledge for Interpretable and Competitive Multi-class Human Activity Recognition" reports results on various different classification architectures used to solve a multi-class problem in the area of human activity recognition, from data coming from one magneto-inertial measuring unit. The research objective is of interest for the portion of Sensors readers that are interested in classification techniques more than in the hardware aspects of wearable sensors. I had no difficulties reading the manuscript, which has the following strengths in terms of structure: the survey on the related works is complete and critically analysed; the statement of the research objectives is clear and unambiguous; the description of the methods for multi-class decomposition is useful for readers who are not expert in the field; the methods (architectures and evaluation metrics) are detailed as one would expect, apart from some minor requests listed below). I have no major objections from the publication of the work on sensors, but I would like the authors to address the elements below, before going on with the publication process:

- The dataset is obtained from data coming from 17 individuals, but I can find nor mention on the overall number of "samples" the were used for the evaluation. This needs to be given to the reader.
Were there any precautions made to prevent the IMU sensor from moving with respect to the back-pack? If the feature are taken from each axis independently, movement of the IMU wrt the backpack (especially in high energy activities as running) may result in decreased accuracy, especially for the part regarding orientation wrt gravity.
- I can see no mention on the overall number of features that were fed to the ML techniques. This needs to be added. Plus, it is my understanding that only statistics-based features were used. Please provide the rationale for this choice.
- I guess that the raw data-set includes transitions from on activities from another. Did you leave transitions out of the successive phases? With 3s long windows, I guess you may end up having "samples" for which transitions between activities are present.
- For the training of the different hierarchies and architectures, 10-fold cross-validation is been reported. Was there a criterion for balancing between classes in these folds? Please explain.
- In the discussion section, I would suggest the authors to expand a bit on the comparison between expert hierarchies and classical multiclass architectures.
- For the same reason, I would ask the authors to give some possible insights on the superior performance of GBT as compared to the other techniques. Tis may be of use for the scientific community working on these aspects.

Reviewer 2 Report

In general, a well written paper, although in a "not-so-hot" domain, but still important in terms of research going towards explainability of a machine learning algorithm's output. Analysis of the results is sound. Although the results are quite tedious to read through.

In line 34 you state: "(...) most machine learning classification algorithms are designed to deal with two target classes, such a multi-class problem must first be decomposed into a set of binary problems. (...)" I would not be so absolute about that. It looks like most of today's popular ML methods line ANNs directly output a multi-class vector.

As mentioned above please highlight shortly the contribution of hierarchical models to understanding/explaining why (with respect to the input data and the domain knowledge, if incorporated) a certain classifier has taken a certain decision.

Reviewer 3 Report

The authors present an approach for multi-class decomposition based on expert hierarchies, and apply it to a Human Action Recognition (HAR) dataset.

The authors argue that hierarchical multi-class decomposition is a popular approach for HAR. However, I see them only citing [1] and [9] which utilize such hierarchical decomposition. These two references are from 2006 and 2004, respectively, so almost 15 years old. Recent works such as "Human Action Recognition Using Deep Multilevel Multimodal (M2) Fusion of Depth and Inertial Sensors , Z. Ahmad et al, IEEE Sensors 2019" directly achieves multi-class classification without hierarchical decomposition. I am not familiar with any well-cited recent work that puts forward the argument that hierarchical decomposition is necessary or useful for HAR. The authors need to provide more justification as to why we would need multi-class decomposition for HAR. Also, I understand this is not a deep learning work, but at least a nod to recent success of deep learning in HAR is warranted. The recent switch to deep learning also puts the multi-class decomposition argument in question, since deep learning-based approaches are inherently multi-class.

I was excited by the word "interpretable" in the title, I can see why a hierarchical decomposition could be more interpretable, and I would say the expert hierarchies are the most interesting part of the work. But I don't see any results that justify the use of multi-class decomposition for such hierarchical results. We can simply use a multi-class classifier, achieve the true label, then use the expert hierarchy to communicate the results to a system. Why do we need to do multi-class decomposition? In fact, tying the expert hierarchies to the algorithm itself is detrimental, what if I need to switch my use case from emergency vs non-emergency to stationary vs mobile?

Multi-class decomposition is also computationally expensive, that is another argument against using multi-class decomposition. I would like to see how MCL compares to the other approaches in Table 2 in terms of computational time. If MCL is significantly faster, why bother with multi-class decomposition?

The authors argue that SVM is not multi-class. However, there are many well-cited works that solve a single optimization problem for a true multi-class SVM, rather than decomposing it into multiple binary classification problems. These works date back to the popular work by Weston and Watkins, Multi-class Support Vector Machines, 1998. Authors should add true multi-class classification results for SVM as well.

Finally, drawing such elaborate analysis and conclusions (section 5 and 6) from results on a single dataset is not acceptable. Many similar datasets are publicly available, such as Heterogeneity HAR dataset, the OPPORTUNITY dataset. Authors should increase the number of datasets rather than providing elaborate analysis on just one dataset. Definitive conclusions cannot be drawn from just one dataset.

Round 2

Reviewer 3 Report

The authors have provided reasonable answers to my queries.